# Nutritional Content and Microbial Load of Fresh Liang, *Gnetum gnemon* var. *tenerum* Leaves

**DOI:** 10.3390/foods12203848

**Published:** 2023-10-20

**Authors:** Sunisa Siripongvutikorn, Worapong Usawakesmanee, Supachai Pisuchpen, Nicha Khatcharin, Chanonkarn Rujirapong

**Affiliations:** 1Centre of Excellence in Functional Foods and Gastronomy, Faculty of Agro-Industry Prince of Songkla University, Hat Yai, Songkhla 90110, Thailand; worapong.u@psu.ac.th (W.U.); jasmeen.vy@gmail.com (N.K.); chanonkarnrujirapong@gmail.com (C.R.); 2Centre of Excellence in Bio-Based Materials and Packaging Innovation, Faculty of Agro-Industry Prince of Songkla University, Hat Yai, Songkhla 90110, Thailand; supachai.p@psu.ac.th

**Keywords:** *Gnetum gnemon*, nutritional, preparation, quality, stem, washing

## Abstract

Liang (*Gnetum gnemon* var. *tenerum*) leaves are widely consumed as a green vegetable in Southern Thailand, and the plant is valued for its nutritional benefits. However, like other leafy greens, liang is vulnerable to microbial contamination, generating foodborne illnesses. This study examined the nutritional content and microbial load of liang leaves at different maturity stages and the effects of washing with chlorinated water. Various growth stages were analysed for proximate composition, amino acids, vitamins, and minerals. Results revealed distinct nutritional profiles, with tip leaves rich in protein and fat and intermediate leaves high in dietary fibre. Liang leaves are abundant in essential amino acids and proteins. Washing with chlorinated water increased leaf weight due to water retention but also caused physical damage, fostering microbial growth and spoilage. Microbiological analysis showed marginal reductions in total viable counts after washing with chlorinated water and significant decreases in coliform and *Escherichia coli* counts. However, stem detachment during washing increased the coliform and *E. coli* counts. Liang leaves exhibited favourable nutritional content, especially in the intermediate stage. Proper handling and storage of liang leaves are crucial to preventing physical damage and microbial contamination. Improved food safety measures, including appropriate post-harvest washing and handling of leafy vegetables, will ensure that consumers can safely enjoy the nutritional benefits of liang leaves.

## 1. Introduction

Liang (*Gnetum gnemon* var. *tenerum*) leaves are a popular leafy vegetable in southern Thailand. Liang belongs to the *Gnetum* family and differs from its tree-type counterparts in Indonesia and Malaysia [1]. Liang is a primitive plant found in southern Thailand and its neighbours under different names and characteristics as well as varieties. In Thailand, liang, *Gnetum gnemon* var. *tenerum*, is a shrub plant, while *G. gnemon* var. *gnemon* is a tree found in other Pacific regions such as Indonesia, Malaysia, and Australia [1]. In var. *tenerum*, the leaves are mainly used, while in var. *gnemon*, the wood and seeds are also harvested. Liang can be cultivated all year round and is resilient to seasonal changes. Recently, the nutritional and health characteristics and outbreaks of human infections associated with the consumption of fresh or minimally processed fruits and vegetables have received increasing attention. Liang leaves contain fibre and proteins with complete amino acids, chlorophyll, and phenolic compounds, which have antioxidant and antimicrobial properties that promote the growth of gut microbiota [1,2]. Anisong et al. [1] reported that the major nutrients in dried liang leaves were carbohydrates (56.8%), protein (27%), and dietary fibre (36.3%).

At present, consumption of fresh produce, particularly minimally processed vegetables, has increased, mainly because of changes in lifestyle and the drive towards convenience, with less time spent on preparing food. The contamination of leafy vegetables via pathogenic microorganisms poses a serious health threat to consumers. Leafy greens and fresh salad vegetables have caused outbreaks of foodborne illnesses due to contamination with human pathogens such as *Escherichia coli* O157:H7, *Listeria* spp., and *Salmonella* spp. [3]. Between 2000 and 2016, several hundred affected individuals and some deaths were directly caused by contaminated leafy vegetable consumption [4]. Freshly consumed produce can be contaminated with pathogens via exposure to contamination sources somewhere from farm production to the point of sale in the market. Normally, on farms, the tip, young, and intermediate stages of liang leaves are reaped, and the stems of leaves are tied, then sprayed or dipped in water and sent to the market. Potential microbial contamination sources in the supply chain include soil, manure, faeces, dust, unhygienic water, and handling during the pre- or post-harvest stages. In fields on farms, the use of untreated water for irrigation can lead to a high risk of contamination with human enteric bacteria. Untreated manure, dirty farm utensils and unhygienic washing water are risk factors for human pathogens (Mogren et al., 2018) [4]. 

Washing is necessary in processing plants and packinghouses for almost all fresh produce to remove dirt, debris, and microorganisms [5], retain weight, and reduce the heat of fresh produce. Farmers normally wash their fresh produce with tap water or chlorinated water, but washing water is usually reused and recycled, leading to the risk of cross-contamination via pathogenic microorganisms between different batches, while washing water may also act as a source of human contamination [6,7]. Published data on the nutritional content and microbial load of liang leaves in relation to a post-harvest method are scarce, while the nutritional content of leaves at various maturity stages and stems (by-products) has not been well documented. Therefore, this study investigated whether washing in chlorinated water impacted the nutritional value and microbial load of liang leaves, following the guidelines of commercial practice to improve food safety measures.

## 2. Materials and Methods

### 2.1. Chemicals and Reagents

The main chemicals, namely methanol, hexane, acetone, acetonitrile, water and glacial acetic acid, were HPLC grade and purchased from RCI Labscan, Bangkok, Thailand. Heptafluorobutyric acid and O-phosphoric acid were purchased from Pierce (Dallas, TX, USA) and Loba Chemie (Mumbai, India), respectively. Other chemicals were AR grade. L-ascorbic acid (Merck, Darmstadt, Germany) and hippuric acid (Superco, St. Louis, MI, USA) met analytical standards. Enzymes, including protease, amyloglucosidase and α-amylase, were purchased from Sigma. The sodium hypochlorite used was commercial grade.

### 2.2. Leaf Preparation and Sampling

Liang leaves used in this study were bought from a farm in Songkhla Province, Thailand, with water irrigation sources from a well and a canal (Figure 1). Samples at the edible stage, including the tip, young, and intermediate leaves, were manually picked from selected fields, purchased, and transferred to the laboratory within 24 h. Leaf tips aged no more than 7 days are always brownish-red after shooting, while young leaves are brownish-red or slightly brownish-red, turning light green between 7 and 14 days of age, and intermediate leaves turn light green between 14 and 25 days of age (Figure 2). Random sampling was applied to the collected leaves and stems (by-products) to determine the nutritional content as proximate compositions, dietary fibre, amino acid profiles, vitamins and minerals. Fresh liang leaves at the young and intermediate stages (pae-salat) were detached from the stem using sanitation gloves. The samples were washed with 100 ppm chlorinated water by soaking at a ratio of 1:2 for 10 min before rinsing twice with tap water to ensure that the free chlorine residue did not exceed 1 ppm. After washing, the excess water was drained in a basket for 10 min with a controlled leaf pile thickness of not more than 1 cm. Thereafter, the samples were divided into four groups: no-washing with stem (NWS), no-washing without stem (NWNS), washing with stem (WS), and washing without stem (WNS) before storing at 4 °C for 12 days. Weight change and physical damage to the leaves, such as bruising, tearing and other abnormal appearances, were recorded, while increases in microbial load were observed as total viable count (TVC) of coliforms, *E*. *coli*, *Salmonella* spp., *Staphylococcus aureus*, *Bacillus cereus*, yeast and mould.

### 2.3. Physiochemical and Chemical Characteristics

#### 2.3.1. Proximate Compositions

Proximate compositions, including moisture, carbohydrate, protein, fat, ash, total dietary fibre, total insoluble and total soluble dietary fibre, were evaluated by the Central Laboratory (Thailand) Co., Ltd. (Songkhla, Thailand) (ISO/IEC 17025:2017). These values were determined as follows.

##### Moisture Content [8]

Moisture content was analysed according to method 950.6 (B). Two grams of sample were dried in an aluminium can (dried and weighed) in a hot air oven (FED 115, Binder, Bohemia, NY, USA) at 100–102 °C for 16–18 h and then cooled in a desiccator and weighed. Thereafter, the sample was redried for 2–4 h and reweighed to ensure constant sample weight.

##### Carbohydrate Content [9]

Carbohydrate content was calculated as Equation (1).
(1)Total carbohydrates (%)=100−(protein+fat+moisture+ash) 

##### Crude Protein Content

Protein determination was analysed using the Kjeldahl method according to AOAC [8] method 981.10. Briefly, 2 g of ground sample was weighed, added onto Whatman 541 and then folded. The sample was transferred to a 250 mL digestion tube and added with 2–3 boiling chips, two catalyst tablets, 15 mL H_2_SO_4_ and 3 mL 30–35% H_2_O_2_. Sample tubes were heated at 410 °C in an auto digestor (Tecator 2508, Foss, Hillerød, Denmark) until the sample became clear (approximately 45 min). After cooling for 10 min, the mixture was steam distilled with (Kjeltec™ 8200, Foss, Hillerød, Denmark) into a receiving flask containing 25 mL H_3_BO_3_ solution with a mixed indicator. Steam distillation was performed until 100-125 mL of green solution was collected in a receiving flask. The solution was then titrated with 0.2 M HCl to neutral grey, and protein content was calculated following Equation (2).
(2)$$\mathrm{Protein}\;\left(\%\right)=\left(V_{A}-V_{B}\right)\times 1.4007\times M\times 6.25\;$$
where V_A_ and V_B_ = volume of HCl used in sample and blank, respectively
1.4007 = milliequivalent weight N × 100 (%)M = molarity of HCl6.25 = protein factor for meat products (16% N).

##### Fat Content

Fat content was analysed using the acid hydrolysis method (Mojonnier analyser, EXE1809-04, Moplant, Geldermalsen, The Netherlands) according to AOAC [8] method 948.15. Briefly, 8 g of the sample was weighed into a beaker, and 2 mL of HCl was added. The mixture was well homogenised (T18 Digital, IKA, Staufen, Germany); thereafter 6 mL HCl was added and heated using a water bath for 90 min with constant stirring. After cooling, the mixture was added to an extraction flask with petroleum ether. After extraction, the sample was dried until it obtained constant weight.

##### Ash Content

Ash content was determined according to AOAC [8] method 923.03. Briefly, 3–5 g of sample was placed into an ash dish that had been already heated, cooled and weighed. The sample was heated at 550 °C using an Isotemp 650 Series Model 58 (Fisher Scientific, Waltham, MA, USA) until light grey ash was obtained and then cooled and weighed. The sample was then reheated and weighed until it obtained constant weight. Ash content was calculated using Equation (3).
(3)$$\mathrm{Ash}\;\mathrm{content}\;\left(\%\right)=\frac{{(W}_{3}-W_{2})\times 100}{W_{1}}\;$$
where
W_1_ = weight of sample (g)W_2_ = weight of crucible (g)W_3_ = weight of crucible + ash (g).

##### Total Dietary Fibre

Total dietary fibre was determined according to AOAC [8] method 985.29 (Enzymatic-Gravimetric Method). Briefly, 1 g of sample was weighed and mixed with 50 mL pH 6.0 phosphate buffer. The pH of the solution was checked and adjusted to 6.0 ± 0.2. Then, 0.1 mL of Termamyl solution was added and heated at 96–100 °C in a water bath (WNB 14, Memmert, Schwabach, Germany) for 15 min with shaking every 5 min. The solution was then cooled to room temperature, and the pH was adjusted to 7.5 ± 0.2 by adding 0.275 M NaOH. Then, 0.1 mL of 50 mg protease in 1 mL phosphate buffer was added to the solution. The mixture was incubated at 60 °C for 30 min with continuous agitation, cooled, and then added to 10 mL of 0.325 M HCl, and the pH was adjusted to 4.0–4.6. Amyloglucosidase 0.3 mL was added, and the mixture was incubated at 60 °C for 30 min before adding 280 mL of 95% ethyl alcohol and preheating to 60 °C to allow the precipitate to form at room temperature for 60 min before filtrating to obtain the residue as soluble and insoluble fibre. The residue was used to determine protein and ash contents. Total dietary fibre (TDF) was calculated as Equation (4), while the blank was calculated as Equation (5).
(4)$$\mathrm{TDF}=[\frac{\mathrm{weight}\;\mathrm{residue}\;-P-A-B}{\mathrm{weight}\;\mathrm{test}\;\mathrm{portion}}]\times 100\;$$
where weight residue = average of weights (mg) for duplicate blank determinations and
P and A = weights (mg) of protein and ash, respectively.Weight test portion = average of 2 duplicate weights (mg) andB = blank.

(5)$$\mathrm{Blank}\;\left(\mathrm{mg}\right)=\mathrm{weight}\;\mathrm{residue}-P_{B}-A_{B}$$
where weight residue = average of weights (mg) for duplicate blank determinations and P_B_ and A_B_ = weights (mg) of protein and ash of blank residues, respectively.

##### Insoluble and Soluble Dietary Fibre

Insoluble and soluble dietary fibre was determined according to AOAC [8] method 991.43. A 1 g sample was mixed with 40 mL of MES-TRIS buffer (pH 8.2). Then, 50 µL of stable heated α-amylase was added to the mixture and stirred at low speed. The mixture was incubated at 95–100 °C in a water bath for 15 min with continuous stirring. After cooling to 60 °C, the sample was added with protease solution and then incubated for 30 min at 60 °C with continuous stirring. Then, 5 mL of 0.561M HCl was added to the sample and the pH was adjusted to 4.0–4.7 using 1M NaOH or HCl solution. Thereafter, 300 µL of amyloglucosidase solution was added and incubated at 60 °C for 30 min with continuous stirring to obtain a sample mixture for insoluble and soluble dietary fibre determination. The sample mixture was filtered into a filtration flash, and the residue was washed with 10 mL of water at 70 °C. The filtrate was used to determine soluble fibre, while the residue was used to determine insoluble fibre. The residue was washed twice with 15 mL of 78% and 95% ethanol and acetone. After washing, the residue was dried overnight at 105 °C in an oven, then cooled and weighed. Insoluble dietary fibre (IDF) and soluble dietary fibre (SDF) were calculated using Equation (6). The blank of this method was calculated as Equation (7).
(6)$$B=\frac{BR_{1}+BR_{2}}{2}-P_{B}-A_{B}$$
where
BR_1_ and BR_2_ = residue weights (mg) for blankP_B_ and A_B_ = weights (mg) of protein and ash of blank.

(7)$$\mathrm{IDF}\;\mathrm{or}\;\mathrm{SDF}=\frac{{(R}_{1}+R_{2})/2-P-A-B}{(M_{1}+M_{2})/2}\times 100\;$$
where
R_1_ and R_2_ = residue weights (mg) for duplicate samplesP and A = weights (mg) of protein and ash, respectivelyB = blank weight (mg)M_1_ and M_2_ = weights (mg) of sample.

#### 2.3.2. Amino Acid Profiles

Amino acid profiles were determined by the Central Laboratory (Thailand) Co., Ltd. (ISO/IEC 17025:2017) according to the LC-MS of Sarwar et al. [10]. Briefly, hydrolysate was prepared by hydrolysing the sample for 22 h with 6N HCl at 110 °C. Amino acids such as methionine, cysteic acid, lysine and tryptophan were prepared by performic acid oxidation of the protein before hydrolysis with 6N HCl. Lysine was determined using sodium borohydride before hydrolysing with HCl. For tryptophan determination, the sample was hydrolysed with 4.2N NaOH and centrifuged for 10 min at 3000× *g*. The supernatant was then injected into the column (Waters Bondapak C18, 30 cm) with a detection range set as 0.2 AUFS and 280 nm. Other sample hydrolysates were prepared by diluting with 710 mg disodium hydrogen phosphate in 1 L water-acetonitrile (95:5) and adjusting the pH to 7.40 using phosphoric acid. All dilute hydrolysates and amino acid standards were evaporated at 35 °C under nitrogen for 15 min or until dry. Thereafter, the dried mixture was mixed with methanol and redried at 35 °C for 15 min before use. 

Preparation and detection using HPLC technique. (1260 Infinity II LC System, Agilent, Santa Clara, CA, USA) 

Solvent A was prepared by mixing 11.45 g sodium acetate in 900 mL water with 46.5 g acetonitrile and 0.5 mL TEA and then diluted to obtain 1 L of solvent. Solvent B was prepared by adding water (410 g) to 475 g acetonitrile. Elution was started at 1 mL/min and held at 100% solvent A for 0.5 min. A convex curve was used to increase solvent B to 46% over the next 9.5 min, then increased to 100% over 0.5 min and maintained for 1.5 min. The column (Waters Pico-Tag, 15 cm) was then equilibrated with solvent A at 1.5 mL/min for 7.5 min with a total analysis time of 20 min. Solvent A (0.01 M sodium acetate and 10% acetonitrile with pH 5.8 adjusted with glacial acetic acid) and solvent B (acetonitrile mixed with water at a ratio 60:40) determination of tryptophan were different from solvent A and B for other amino acids. Elution was started at 1 mL/min with 100% solvent A for 10 min. The column was then washed with solvent B for 2 min and equilibrated with solvent A at 1.5 mL/min for 10 min, with a total analysis time of 22 min.

#### 2.3.3. Vitamin and Mineral Contents

Vitamins A, B_9_ and C and minerals including calcium, copper, iron, magnesium and zinc were evaluated by the Central Laboratory (Thailand) Co., Ltd. (ISO/IEC 17025:2017).

##### Vitamin A

Vitamin A content was determined as beta-carotene according to Visuthi [11] with modification using HPLC instead of a spectrophotometer. Briefly, the sample was ground with anhydrous Na_2_SO_4_ (1:1) and then repeatedly extracted with acetone. All extracts were combined and concentrated under a vacuum. Carotenoid pigments were then transferred into diethyl ether by distilled water addition. The pigment in the aqueous layer was repeatedly extracted. The combined ethereal layers were rehydrated over anhydrous Na_2_SO_4_, then evaporated (V-700 Vacuum Pump, Buchi, Bangkok, Thailand) and completely dried under reduced pressure. Benzene was mixed with the residue and then analysed using HPLC with column mirror/glass (Phenomenex, 15 cm × 4.6 mm, 3-micron, silica (2) 100A), using hexane-acetone grade HPLC (87:13) as the mobile phase with flow rate 1.4 mL/min for 20 min under wavelengths 450 nm and 470 nm.

##### Vitamin B_9_ [12]

The sample was mixed with an internal standard (hippuric acid solution), water and ammonia and then shaken for 1 min. After shaking, the mixture was extracted in an ultrasonic bath (RK 100 H, Babdelin, Berlin, Germany) for 30 min at 60 °C, cooled, and the pH was adjusted to 7 using formic acid. Thereafter, the mixture volume was adjusted to 100 mL before filtering through a 0.45 µm nylon membrane and 5 µL was injected into an HPLC system with electrospray ionisation (ESI) (Agilent G1948B, Agilent Technologies, Santa Clara, CA, USA) using 5 µm C18 silica (Johnson, Dalian, China). An aqueous solution of 5 mM heptafluorobutyric acid was used as mobile phase A while methanol was used as mobile phase B. Flow rate was 1.0 mL/min with temperature maintained at 30 °C. Positive and negative modes of electrospray ionisation were operated and continuously switched with the voltage of capillary, extractor, and RF lens at 3.2 kV, 4 V and 0.5 V, respectively; 110 and 250 °C were used for the source and desolvation temperatures while gas flow for desolvation and cone was set at 300 and 50l/hour. Full scan mass spectra were acquired from *m*/*z* 100 to 500.

##### Vitamin C

Vitamin C was determined according to the method of the National Bureau of Agricultural Commodity and Food Standards [13]. A standard curve was prepared using L-ascorbic acid with concentrations 0.25, 0.5, 10, 20 and 30 µg/mL to obtain peak area. The sample was mixed with 3% m-phosphoric acid, then shaken and sonicated in an ultrasonic bath. The sample was then filtered and injected into an HPLC with column (5 µm Lichrospher, 125 × 4 mm, Merck) with eluent flow rate 0.5 mL/min at ambient temperature (Mobile phase: KH_2_PO_4_ in o-phosphoric acid).

##### Mineral Determination

Mineral determination followed AOAC [8] method 984.27. A 1.5 g aliquot of dried sample was mixed with 30 mL HNO_3_·HClO_4_ (2 + 1) and left to sit overnight. The Kjeldahl flask with the sample was heated on a mantle at low temperature with continuous heating at low temperature until HNO_3_ and H_2_O were vaporised. The flask was then switched to a cool heating mantle with occasional heating until digestion was completed. The reaction of HClO_4_ and organic material was stopped when the effervescent reaction ended. Thereafter, the mixture was heated at a high temperature for 2 min and then cooled. The mixture, with the final acid content (HClO_4_) at 20%, was diluted with water and left overnight. Minerals in the mixture were determined using inductively coupled plasma (ICP) emission spectroscopy (BRE731400 iCAP PRO ICP-OES, Thermo Scientific, Waltham, MA, USA) with wavelengths for calcium, copper, iron, magnesium and zinc as 317.9, 324.7, 259.9, 383.2 and 213.8 nm, respectively and then calculated as Equation (8).
(8)C=A×(50/B)
where
A = concentration (µg/mL) of element as determined using ICPB = volume or weight of sample as ml or gC = elemental concentration in the sample solution (µg/mL or µg/g).

### 2.4. Weight Gain and Loss

The sample weight was recorded daily for weight gain and loss by comparison with the initial weight, calculated using Equation (9).
(9)$$\mathrm{Weight}\;\mathrm{gain}\;(\%)=\frac{\left(W_{1}-W_{2}\right)}{W_{1}}\times 100\;$$
where
W_1_ = initial weight of the sampleW_2_ = weight of the sample after storage.

### 2.5. Damage Detection and Classification

The damage level was detected and quantified following Mulaosmanovic et al. [14], with some modifications, by comparing photographs of healthy and incomplete leaves. 

### 2.6. Microbial Load

The microbial load was evaluated by determining total viable count (TVC), coliform bacteria, *Escherichia coli*, yeast and mould at the Centre of Measurement and Standard Accreditation, Faculty of Science, Prince of Songkla University (ISO/IEC 17025:2017).

#### 2.6.1. TVC Determination

TVC was analysed according to the method of Maturin and Peeler [15]. Briefly, the sample and Butterfield’s phosphate buffer were well mixed. The mixture was added to a blender jar and blended to obtain a 10^−1^ dilution. Then, appropriate dilution was made. One millilitre of each dilution was transferred to a plate, and 15 mL of plate count agar was added. The plates were rotated to spread and solidify the agar, then inverted and incubated at 35 °C for 48 h. The number of microbes was counted and recorded in colony-forming units per gram (log CFU/g).

#### 2.6.2. Coliform and *E. coli* Determination

Coliforms and *E. coli* bacteria were analysed according to the method of Feng et al. [16]. The sample was prepared following the method used for TVC. Each dilution was transferred to 3 lactose broth tubes. All the tubes were incubated at 35 °C for 24 h. The tube containing gas was recorded and further incubated for another 24 h to record gas production. One loop-full of lactose broth tubes was transferred to a brilliant green lactose bile (BGLB) broth tube. The tubes were incubated at 35 °C for 48 h. All tubes containing active gas were recorded to calculate the most probable number (MPN). To confirm *E. coli* in the sample, a loopful from gassing lactose broth tubes was transferred to an EC broth tube and incubated for 24 ± 2 h at 44.5 °C. After incubation, the tube was examined for gas production. If negative, the tube was re-incubated and examined again at 48 ± 2 h. A loop-full of gassed EC broth was streaked for isolation on an L-EMB agar plate and incubated for 18–24 h at 35 °C. A colony dark-centred and flat, with or without metallic sheen, was selected for testing the gram strain, indole production, Voges-Proskauer (VP)-reactive compounds, Methyl red-reactive compounds, citrate and gas from lactose for confirmation.

#### 2.6.3. Yeast and Mould Determination

Yeast and mould were analysed according to the method of Tournas et al. [17]. Samples in 0.1% peptone water were mixed to make 10^−1^ dilution. The mixture was homogenised, and appropriate dilutions were made. One millilitre of each sample dilution was transferred onto the plates, and Dichloran rose bengal chloramphenicol (DRBC) agar was added. The plates were mixed by swirling clockwise and counterclockwise, then incubated in the dark at 25 °C. Incubations containing 10–150 colonies were selected for counting, and the number of microbials was recorded as log CFU/g.

### 2.7. Statistical Analysis

All experiments were performed in triplicate, and all quality parameters were assessed using a completely randomised design (CRD). Differences in mean values were tested using ANOVA, with specific differences between groups or treatments assessed using Tukey’s test at the 5% level of significance. The experiments were performed in triplicate, with each analysis in duplicate.

## 3. Results and Discussion

### 3.1. Proximate Compositions

Moisture, carbohydrate, insoluble dietary and soluble dietary fibre contents in the leaves increased with leaf age as expected. The highest values were found in the stems, followed by the intermediate leaves (Table 1). The highest levels of protein, fat and ash were found in the tip leaves and decreased with age. A decrease in protein content with leaf maturity was also reported in Moringa leaves [18], while Ncube et al. [19] found that protein content decreased while fibre content increased in aged leaves. Reduction in protein content with leaf ageing was caused by an increase in dietary fibre. Increasing fibre content in leaves leads to greater stiffness and toughness to withstand the optimal angle for light reception and provides more resistance against external forces such as wind and animals to ensure maximum leaf life span [20]. Protein and fibre contents in the intermediate stage of liang leaves were higher than in leafy vegetables [21], whereas carbohydrate contents were similar. Fat content in the intermediate stage of liang was close to values reported in *A. digitata* and *A. hybridus* but lower than in *H. sabdariffa* and *V. unguiculata* [21]. Liang leaves contained carbohydrate, protein and fat content close to spinach at 29.72 ± 0.51, 29.36 ± 0.49 and 2.80 ± 0.11 g DW, chard 33.49 ± 1.18, 28.38 ± 0.04 and 2.68 ± 0.12 g DW and parsley 24.85 ± 0.60, 25.13 ± 0.06 and 3.37 ± 0.09 g DW, respectively but contained less ash content than these three vegetables [22]. Dietary fibre in liang leaves, especially in the intermediate stage, was higher than in peas, corn, carrots, leeks, spinach, chard, coriander and parsley [22]. Results revealed that the nutrition of liang leaves was comparable to other leafy vegetables and classified as high fibre.

### 3.2. Amino Acid Profile

Amino acid contents in the various growth stages of liang leaves and stems are listed in Table 2. The highest contents of both essential and non-essential amino acids were found in the tips of liang leaves and decreased with leaf maturity, with the lowest amino acid contents recorded in the stems. A decrease in amino acids from young to mature leaves was also reported in mustard and peach leaves [24], but no significant differences were found in the amino acid contents of young and mature *Sporobolus stapfianus* leaves [24]. Results indicated that changes in amino acids between young and mature leaves were species-dependent.

Results indicated that liang leaves at all growth stages contained complete essential amino acid contents similar to leek, spinach, chard, parsley and coriander [22], while the leaves contained higher arginine than leek, spinach, chard, parsley and coriander at 2.88 ± 0.21, 3.75 ± 0.19, 2.47 ± 0.17, 2.69 ± 0.19 and 1.89 ± 0.11 g/100 g DW, respectively [22].

### 3.3. Vitamins and Minerals

Higher levels of vitamin A and calcium were found with increased leaf maturity, whereas vitamin C, iron, zinc and magnesium levels decreased (Table 3). Interestingly, folic acid was only found in the tip and young leaves and was absent in the intermediate and old leaves. Generally, the role of folic acid in plants controls the development, signalling cascading and metabolism of carbon and nitrogen [25]. Folic acid also helps cell division but inhibits elongation [26]. Folate content decreased by up to 40% depending on vegetable species after storage at 22 °C for 2 h, while decreasing folate content was found in injured plants [27]. The liang leaves were harvested every 2 weeks, and folic acid in this plant was utilised during budding and tipping production (apex or young leaves). By contrast, Rubóczk and Hájos [28] reported that the folic acid content of 30-day-old fresh beetroot leaves ranged from 58.77–111.08/100 g and lower than 75.4–113.86 μg/100 g for 50-day-old leaves. 

Carotenoids are used as stabilisers in pigment-protein complexes, harvesting energy from sunlight and protecting plant cells from light (photoprotection), especially β-carotene, the main carotenoid in plants [29]. Cheptoo et al. [30] and El-Nakhel et al. [31] reported that mature vegetables contain higher levels of β-carotene than young or microgreen vegetables. Increasing the light absorption area of mature vegetables led to an increase in the need for antioxidant compounds for photoprotection and light harvesting [31]. Reduction in ascorbic acid with vegetable maturity was found in green butterhead lettuce, but ascorbic acid content in red butterhead lettuce increased with maturity [31]. Ascorbic acid is normally used to protect young leaves from light damage caused by radicals, particularly reactive oxygen species (ROS) [31], while a decrease in zinc and an increase in calcium with leaf age were found in Moringa leaves [18]. Plants use zinc using various mechanisms such as enzyme components, disease resistance, photosynthesis, cell structure, protein synthesis, pollen formation, antioxidant mechanisms and chlorophyll formation [32], whereas iron plays an important role in photosynthesis, chlorophyll and DNA synthesis and respiration [33,34]. Decreases in zinc and iron contents with leaf maturity are caused by high metabolic needs in young leaves for protein synthesis and the formation of new leaf cells compared with older leaves. Magnesium is required by plants for enzyme activation, photosynthesis and protein synthesis as an essential component of chlorophyll [35,36]. Interestingly, old leaves of *Camellia sinensis* (L.) Kuntze contained higher magnesium and calcium than young leaves [37]. A decrease in magnesium content in older leaves may be caused by the transfer of magnesium from older leaves to young leaves. Hauer-Jákli and Tränkner [38] reported that plants use older leaves as a magnesium source for young leaves because the growing parts or young leaves require high amounts of magnesium. Plants use calcium as a component of cell walls and membranes to improve rigidity and as a messenger in signalling pathways [39]. An increase in calcium may be related to increased stiffness to maximise leaf life span and angle for light absorption. Minerals and vitamins contained in stems were lower than those in all growing stages of leaves, indicating the lower enzymatic and metabolic activity needed in this plant part.

### 3.4. Weight Changes and Damage Levels from Field to Bagging and Storage 

Weight gain after washing was 42.13 to 86.67% (Figure 3), depending on the stem detaching step. This was further confirmed by an increase in water activity (a_w_) from 0.995 to 0.998 when water was added to the fresh sample at 44.44% of the sample weight (unpublished data). High moisture content and a_w_ lead to both biochemical reactions and microbial growth. In general, a_w_ values above 0.950 are sufficient to support the growth of bacteria, yeast and mould [40]. The washing step in the detached sample resulted in greater weight gain compared with the whole leaf without stem detachment. The stem of the leaf possibly acted as an internal sieve layer during the washing and draining steps, allowing water to contact and be removed more quickly. Without a stem, liang leaves attach to each other because of the surface tension of water, making draining difficult. Thus, washing leafy vegetables with stems improves drainage, with better control of weight gain from water retention.

Liang leaf damage was estimated at 1.53% due to manual picking, collecting and packing in a bag weighing 2 kg. After storage for 8 days, the damage to untreated liang leaves (NWS) increased to 11.92%. However, stem detachment and washing caused a significant increase in damaged leaves to 20.07% and 58.47%, respectively, as shown in Figure 4 and Figure 5. The increase in damage caused by stem detachment and washing steps was also confirmed using the highest damage found in liang leaves (WNS) at 68.87%.

The wilt symptom ratio was more pronounced in unwashed leaves, while rotten characteristics with a specific acid-like smell, such as sweet fermented sticky rice, Thai name “Khao-Mak”, with higher acidity and an unpleasant smell was found in washed samples as storage time increased. Damaged leaves were more pronounced when the samples were prepared from stem detaching and washing steps. Detaching leaves with manual force resulted in a wound at the leaf node and cellular softening at the petiole and leaf plate area, as shown in Figure 4b,d. Transfer of heat from human hands (normally 36.5 ± 0.7 °C) while touching the sample without wearing gloves can increase the temperature of the sample by 0.1–1 °C depending on the holding time (unpublished data), as an additional factor stimulating microbial growth, changing the bacterial type and facilitating pathogenic cross-contamination as well as inducing biochemical reactions. Physically injured cells are prone to increased microbial infestation and cell lesions, with enhanced respiration rates due to higher surface area and nutrient release. Therefore, every processing step involving force application and the addition of water generates increased cell injury. In modern markets or convenience stores, most vegetables are already bagged or packaged to facilitate consumer purchasing and reduce selection time and touching, while in fresh markets, people usually select the produce manually. Marketing surveys and personal interviews determined that fresh produce was 20 to 60% damaged during customer purchasing, owing to the picking and selection process. Therefore, touching is generally prohibited for all highly-priced products. Various fresh produce such as lansium (*Lansium parasiticum*), durian, onion and shallot, as well as cilantro and delicate leafy vegetables, are not washed with water before and after harvesting [41] to better control damage and microbial spoilage during storage, transportation and selling. Damage and spoilage characteristics of different post-harvest steps lead to problems that require further study to better understand and manage the processes.

### 3.5. Microbial Load

After washing with chlorinated water at 100 ppm, results indicated that the total viable count was reduced by only 0.1–1.0 log cfu/g (Figure 6). Other microorganism types were not within the range of export standard regulations (Figure 7) (Table 4 and Table 5). Chlorinated water at 100 ppm did not significantly reduce microorganisms but significantly reduced coliforms and *E. coli*. However, stem detachment resulted in higher coliform and *E. coli* counts in liang leaves without stems (NWNS and WNS). This result concurred with Rosberg et al. [3], who reported that the washing step significantly reduced bacterial load but altered bacterial species composition in leafy vegetables, including spinach (*Spinacia oleracea*) and rocket (*Diplotaxis tenuifolia*). The lower efficacy of chlorinated water for microbial reduction found in this study may be due to the high organic matter content, microbial types, load and leaf architecture. The washing step also led to more excess water, as determined by a gain in weight of up to 86.67%, with an increment in a_w_ matching the gain in weight. The strong relationship between water pick up and a_w_ after washing provided good evidence to support microbial growth. Physical damage such as leaf bruising, tearing and soft rot was also more noticeable, as shown in Figure 4c,d. The preliminary test results showed that liang leaves had high protein content at 4.5 g/100 g fresh leaves or 25.5 g/100 g DW (data in publication). High protein content, low acidity, soft texture and high a_w_ due to the fresh stage and washing step supported microbial growth.

The water used to irrigate liang plants on farms had a high microbial load, TVC (10^6^–10^8^ cfu/mL), with coliforms (>1100 MPN/mL), *E. coli* (less than 100 MPN/mL to 1600 MPN/g), and *Bacillus* spp. (none to more than 1000 cfu/mL), while yeast and mould were also high at 10^4^ to 10^6^ cfu/mL (unpublished data due to government commitment). Farmers generally use reservoirs, streams or natural channels to irrigate their crops with no sanitising treatment; therefore, the risk of contamination by microorganisms is high. Farmers also use their reservoirs for fish culture, and the accumulated feed and faeces are sources of microbial growth. Ali et al. [42] reported that well water used for drinking and washing in Jimma Town, Southwest Ethiopia, had high bacteriological contamination. *E. coli* is widely used as an indicator of faecal contamination in waterways. Kambire et al. [43] stated that water used to irrigate market gardening contained high microorganism loads ranging from 3.64 to 4.35 log cfu/100 mL, 2.44 to 3.31 log cfu/100 mL, 2.44 to 2.9 log cfu/100 mL, and 2.07 to 3.63 log cfu/100 mL for total coliforms, *E. coli*, faecal enterococci and sulphite-reducing clostridia, respectively while mean loadings of mesophilic aerobic bacteria, moulds and yeasts ranged from 4.95 to 5.98 log cfu/100 mL, 1.8 to 2.08 log cfu/100 mL and 1.5 to 1.98 log cfu/100 mL, respectively. Four different mould strains belonging to the genus Aspergillus were also found in the irrigation water. High microbiological contamination in this and previous studies confirmed that water is very important and must have good management for costing and quality balancing.

Several studies have shown that plants harbour microbial contamination, which is harmful to human health. However, fresh produce also contains various endophytes [44,45]. Endophytes or plant probiotic bacteria are naturally occurring plant-associated microorganisms that enhance the growth of host plants, including yield and antioxidant compounds, and may suppress diseases when applied in adequate amounts [46,47,48]. Major genera of plant growth-promoting probiotic bacteria include *Bacillus*, *Paraburkholderia*, *Pseudomonas*, *Acinetobacter*, *Alcaligenes*, *Arthrobacter* and *Serratia* [46,47,49]. The determination of endophytes in both bacteria and fungi after decontamination was similar to the investigation of total viable count (TVC) and yeast and mould in washed produce for export. Therefore, the higher counts found in washed liang leaves could be due to endophytes because the sample was well blended or homogenised to break the plant cells, leading to the liberation of contaminated germs and endophytes. Probiotics in the human body emanate from the regular consumption of fresh vegetables, yoghurt, pickled vegetables and fermented sticky rice. Therefore, it is important to determine the composition of endophytic bacteria, mould, probiotics and pathogens in plant samples or fresh produce. Food safety and quality regulations may need to be reviewed and re-established by separating useful endophytes from harmful pathogens to provide more consumer information and perception. This report is the first to suggest a review of quality regulations and endophytic benefits.

The effectiveness of the washing solution depends on its concentration and chemical contents, as well as the contact time [50]. Recently, chemical rinses, usually called fruit and vegetable washes, have become available to keep fresh produce safe from bacterial contamination. However, the FDA has not yet approved these commercial washes because their residue content and safety have not been evaluated, tested and standardised [51]. Chlorine is not the only washing agent permitted for use in Thailand. Suntornsuk [52] and the Ministry of Public Health [53] reported that washing decontamination chemicals for fresh produce can be bi-sodium carbonate, acetic acid, ozone and calcium hydroxide by soaking for 10–15 min and then rinsing. 

## 4. Conclusions

Liang leaves contain high levels of Mg and are a good source of protein and fibre, but they are unsuitable for consumption because of the high microbial load caused by untreated irrigation water. Using natural groundwater leads to high microbial loads in fresh produce. Washing did not significantly reduce the microbial count, especially after storage at 4 °C for 8 days and caused more bruising and weight gain. The detachment step also increased the risk of bruising but with a lower risk than the washing step. Both washing and detachment steps negatively impacted liang leaves. The most appropriate temperature for storing liang leaves was 4 °C with no preparation steps. The washing agent and decontamination method should be further investigated and improved for microbiological safety purposes. The high nutritional value of liang plant leaves indicates many opportunities for future use.

## Figures and Tables

**Figure 1 foods-12-03848-f001:**
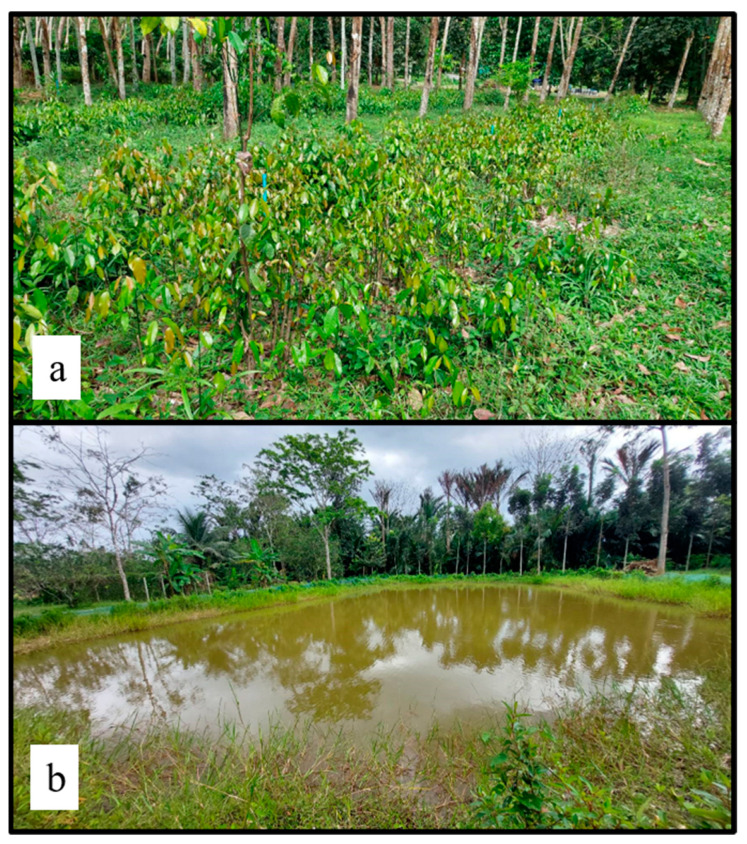
Intercropping of liang shrubs with water irrigation pipe system (**a**) Drilled well for the farm (**b**).

**Figure 2 foods-12-03848-f002:**
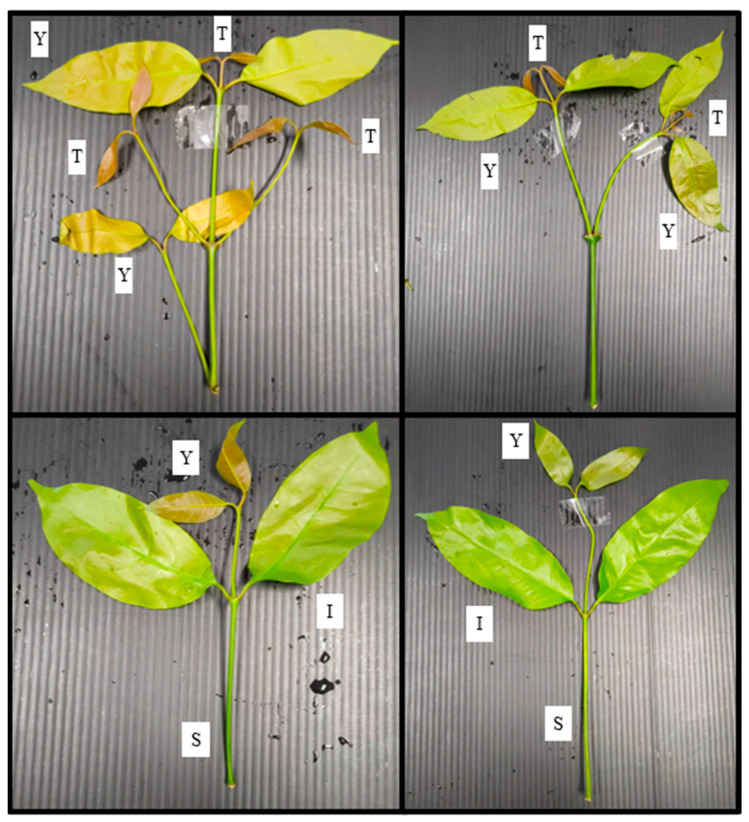
Tip, young and intermediate (pae-salat) stages of liang leaves and stems: T means tip; Y means young; I means intermediate; S means stem.

**Figure 3 foods-12-03848-f003:**
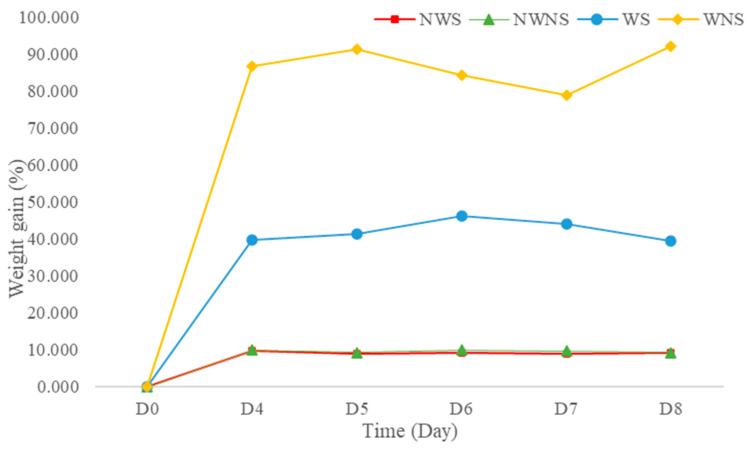
Weight gain of liang (*Gnetum gnemon* var. *tenerum*) leaves after storage at 4 °C for 8 days. NWS means no-washing with stem; NWNS means no-washing without stem; WS means washing with stem; WNS means washing without stem.

**Figure 4 foods-12-03848-f004:**
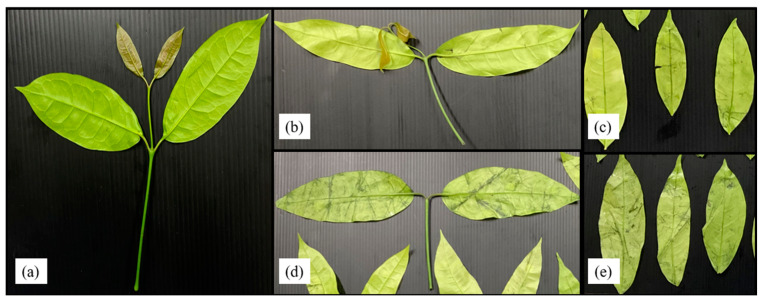
Liang leaves from the field (baseline or initial quality) (**a**); liang leaves after 20 days storage at 4 °C; no-washing with stem (NWS) (**b**); no-washing without stem (NWNS) (**c**); washing with stem (WS) (**d**); washing without stem (WNS) (**e**).

**Figure 5 foods-12-03848-f005:**
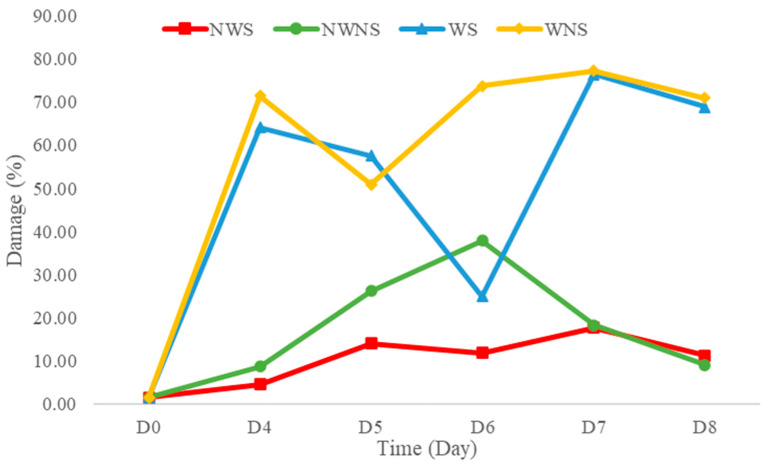
Damage level of liang (*Gnetum gnemon* var. *tenerum*) leaves after storage at 4 °C for 8 days. NWS means no-washing with stem; NWNS means no-washing without stem; WS means washing with stem; WNS means washing without stem.

**Figure 6 foods-12-03848-f006:**
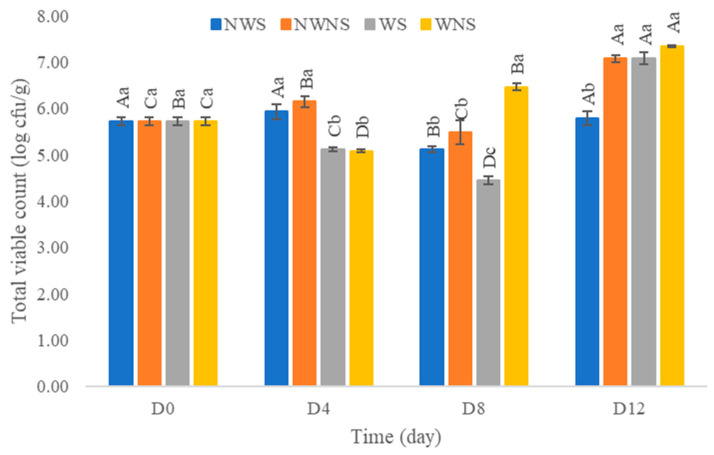
Total viable count of liang (*Gnetum gnemon* var. *tenerum*) leaves after storage at 4 °C for 12 days. Different uppercase letters (A–D) indicate significant differences within the same treatment group. Different lowercase letters (a–c) indicate significant differences between treatment groups within each day (*p* < 0.05). NWS means no-washing with stem; NWNS means no-washing without stem; WS means washing with stem; WNS means washing without stem.

**Figure 7 foods-12-03848-f007:**
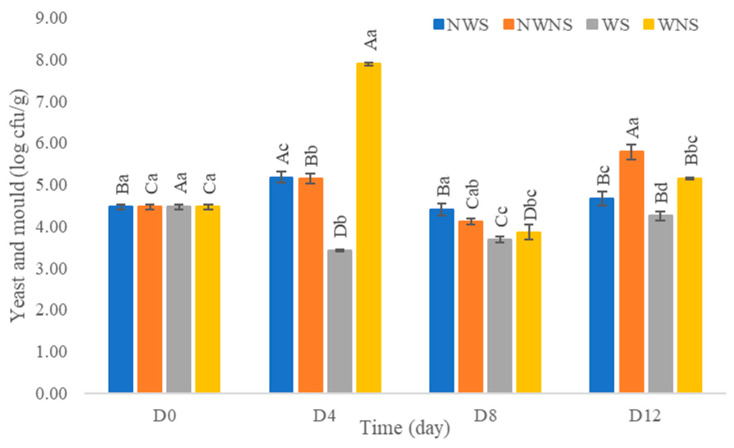
Yeast and mould of liang (*Gnetum gnemon* var. *tenerum*) leaves after storage for 12 days. Different uppercase letters (A–D) indicate significant differences within the same treatment group. Different lowercase (a–c) letters indicate significant differences between treatment groups within each day (*p* < 0.05). NWS means no-washing with stem; NWNS means no-washing without stem; WS means washing with stem; WNS means washing without stem.

**Table 1 foods-12-03848-t001:** Proximate compositions and dietary fibre in tip leaves, young leaves, intermediate leaves and stems of liang (*Gnetum gnemon* var. *tenerum*) (g/100 g DW).

Compound	Treatments				Nutrition Claim Criteria (Wet) *	
	Tip	Young	Intermediate	Stem	Good Source	High
Moisture	82.08 ± 0.29 ^a^	82.93 ± 1.40 ^a^	83.04 ± 1.06 ^a^	82.95 ± 2.08 ^a^	-	-
Carbohydrate	55.13 ± 1.36 ^c^ (9.88 ± 0.24 ^c^)	63.74 ± 2.36 ^b^ (10.88 ± 0.40 ^b^)	65.63 ± 2.63 ^b^ (11.13 ± 0.45 ^b^)	78.42 ± 1.02 ^a^ (13.37 ± 0.17 ^a^)	-	-
Protein	33.43 ± 1.21 ^a^ (5.99 ± 0.22 ^a^)	26.77 ± 0.82 ^b^ (4.57 ± 0.14 ^b^)	25.41 ± 0.68 ^b^ (4.31 ± 0.12 ^b^)	14.25 ± 0.37 ^c^ (2.43 ± 0.06 ^c^)	≥2.50	≥5.00
Fat	4.13 ± 0.15 ^a^ (0.74 ± 0.03 ^a^)	2.69 ± 0.13 ^b^ (0.46 ± 0.02 ^b^)	2.30 ± 0.10 ^c^ (0.39 ± 0.00 ^c^)	1.47 ± 0.02 ^d^ (0.25 ± 0.01 ^d^)	-	-
Ash	7.31 ± 0.06 ^a^ (1.31 ± 0.01 ^a^)	6.80 ± 0.06 ^b^ (1.16 ± 0.01 ^b^)	6.66 ± 0.06 ^b^ (1.13 ± 0.01 ^b^)	5.87 ± 0.16 ^c^ (1.00 ± 0.03 ^c^)	-	-
Total dietary fibre	30.92 ± 0.37 ^d^ (5.54 ± 0.07 ^d^)	37.67 ± 0.21 ^c^ (6.43 ± 0.04 ^c^)	41.27 ± 0.16 ^b^ (7.00 ± 0.03 ^b^)	48.97 ± 0.94 ^a^ (8.35 ± 0.16 ^a^)	≥3.00	≥6.00
Insoluble dietary fibre	27.51 ± 0.20 ^d^ (4.93 ± 0.04 ^d^)	32.22 ± 1.01 ^c^ (5.50 ± 0.17 ^c^)	34.43 ± 0.46 ^b^ (5.84 ± 0.08 ^b^)	41.88 ± 0.42 ^a^ (7.14 ± 0.07 ^a^)	-	-
Soluble dietary fibre	3.40 ± 0.17 ^c^ (0.61 ± 0.03 ^c^)	5.45 ± 0.23 ^b^ (0.93 ± 0.04 ^b^)	6.84 ± 0.29 ^a^ (1.16 ± 0.05 ^a^)	7.10 ± 0.06 ^a^ (1.21 ± 0.01 ^a^)	-	-

Values in parentheses are wet basis; - means no criteria; a–d means different lowercase letters indicate significant differences between treatments (*p* < 0.05). Source (*) Ministry of Public Health [23].

**Table 2 foods-12-03848-t002:** Amino acid contents in tip leaves, young leaves, intermediate leaves and stems of liang (*Gnetum gnemon* var. *tenerum*) (g/100 g DW).

Amino Acids (g/100 g DW)	Treatments			
	Tip	Young	Intermediate	Stem
Essential				
Histidine	0.73 ± 0.028 ^a^ (0.13 ± 0.005 ^a^)	0.70 ± 0.027 ^a^ (0.12 ± 0.005 ^a^)	0.59 ± 0.016 ^b^ (0.10 ± 0.003 ^b^)	0.47 ± 0.016 ^c^ (0.08 ± 0.003 ^c^)
Isoleucine	0.39 ± 0.015 ^a^ (0.07 ± 0.003 ^a^)	0.35 ± 0.010 ^b^ (0.06 ± 0.002 ^b^)	0.12 ± 0.04 ^c^ (0.02 ± 0.001 ^c^)	ND ^d^ (ND ^d^)
Leucine	1.40 ± 0.010 ^a^ (0.25 ± 0.002 ^a^)	1.23 ± 0.015 ^b^ (0.21 ± 0.003 ^b^)	0.88 ± 0.006 ^c^ (0.15 ± 0.001 ^c^)	0.29 ± 0.012 ^d^ (0.05 ± 0.002 ^d^)
Lysine	2.34 ± 0.028 ^a^ (0.42 ± 0.005 ^a^)	1.76 ± 0.059 ^b^ (0.3 ± 0.010 ^b^)	1.30 ± 0.059 ^c^ (0.22 ± 0.010 ^c^)	0.70 ± 0.029 ^d^ (0.12 ± 0.005 ^d^)
Methionine	0.73 ± 0.024 ^a^ (0.13 ± 0.004 ^a^)	0.70 ± 0.006 ^a^ (0.12 ± 0.001 ^b^)	0.59 ± 0.021 ^b^ (0.10 ± 0.004 ^c^)	0.59 ± 0.010 ^b^ (0.10 ± 0.002 ^c^)
Phenylalanine	0.78 ± 0.015 ^a^ (0.14 ± 0.003 ^a^)	0.76 ± 0.010 ^a^ (0.13 ± 0.002 ^b^)	0.59 ± 0.020 ^b^ (0.1 ± 0.003 ^c^)	0.47 ± 0.005 ^c^ (0.08 ± 0.001 ^d^)
Threonine	0.67 ± 0.024 ^a^ (0.12 ± 0.004 ^a^)	0.53 ± 0.015 ^b^ (0.09 ± 0.003 ^b^)	0.35 ± 0.008 ^c^ (0.06 ± 0.001 ^c^)	0.23 ± 0.010 ^d^ (0.04 ± 0.002 ^d^)
Tryptophan	0.39 ± 0.010 ^b^ (0.07 ± 0.002 ^a^)	0.41 ± 0.007 ^a^ (0.07 ± 0.001 ^a^)	0.41 ± 0.003 ^a^ (0.07 ± 0.001 ^a^)	ND ^c^ (ND ^b^)
Valine	1.45 ± 0.010 ^a^ (0.26 ± 0.002 ^a^)	1.35 ± 0.015 ^b^ (0.23 ± 0.003 ^b^)	0.94 ± 0.006 ^c^ (0.16 ± 0.001 ^c^)	0.59 ± 0.026 ^d^ (0.10 ± 0.004 ^d^)
Non-essential				
Alanine	1.90 ± 0.056 ^a^ (0.34 ± 0.010 ^a^)	1.46 ± 0.006 ^b^ (0.25 ± 0.001 ^b^)	0.83 ± 0.016 ^c^ (0.14 ± 0.003 ^c^)	0.41 ± 0.016 ^d^ (0.07 ± 0.003 ^d^)
Arginine	5.58 ± 0.097 ^b^ (1.00 ± 0.017 ^a^)	5.68 ± 0.020 ^b^ (0.97 ± 0.003 ^b^)	5.96 ± 0.010 ^a^ (1.01 ± 0.002 ^a^)	5.92 ± 0.006 ^a^ (1.01 ± 0.001 ^a^)
Cystine	ND	ND	ND	ND
Glutamic acid	2.06 ± 0.015 ^b^ (0.37 ± 0.003 ^b^)	1.29 ± 0.021 ^a^ (0.39 ± 0.004 ^a^)	0.77 ± 0.010 ^c^ (0.13 ± 0.002 ^c^)	0.70 ± 0.026 ^d^ (0.12 ± 0.004 ^d^)
Glutamine	ND	ND	ND	ND
Aspartic acid	3.74 ± 0.148 ^a^ (0.67 ± 0.026 ^a^)	2.28 ± 0.015 ^d^ (0.13 ± 0.003 ^d^)	1.30 ± 0.010 ^b^ (0.22 ± 0.002 ^b^)	1.06 ± 0.027 ^c^ (0.18 ± 0.005 ^c^)
Asparagine	ND	ND	ND	ND
Glycine	0.95 ± 0.015 ^a^ (0.17 ± 0.003 ^a^)	0.76 ± 0.015 ^b^ (0.13 ± 0.003 ^b^)	0.41 ± 0.010 ^c^ (0.07 ± 0.002 ^c^)	0.23 ± 0.006 ^d^ (0.04 ± 0.001 ^d^)
Hydroxylysine	ND	ND	ND	ND
Hydroxyproline	ND	ND	ND	ND
Proline	1.23 ± 0.010 ^a^ (0.22 ± 0.002 ^a^)	1.11 ± 0.041 ^b^ (0.19 ± 0.007 ^b^)	0.65 ± 0.016 ^c^ (0.11 ± 0.003 ^c^)	0.29 ± 0.012 ^d^ (0.05 ± 0.002 ^d^)
Serine	0.95 ± 0.015 ^a^ (0.17 ± 0.003 ^a^)	0.70 ± 0.015 ^b^ (0.12 ± 0.003 ^b^)	0.53 ± 0.016 ^c^ (0.09 ± 0.003 ^c^)	0.47 ± 0.006 ^d^ (0.08 ± 0.001 ^d^)
Tyrosine	0.39 ± 0.015 ^a^ (0.07 ± 0.003 ^a^)	0.41 ± 0.010 ^a^ (0.07 ± 0.002 ^a^)	0.35 ± 0.010 ^b^ (0.06 ± 0.002 ^b^)	0.23 ± 0.011 ^c^ (0.04 ± 0.002 ^c^)
Cysteine	ND	ND	ND	ND
Total essential	8.87 (1.59)	7.79 (1.33)	5.78 (0.98)	3.34 (0.57)
Total non-essential	16.80 (3.01)	13.71 (2.34)	10.79 (1.83)	9.33 (1.59)
Total	25.67 (4.60)	21.50 (3.67)	16.57 (2.81)	12.67 (2.16)

Values in parentheses are wet basis; ND means not detected; - means no specific requirement; a–d means different lowercase letters indicate significant differences between treatments (*p* < 0.05).

**Table 3 foods-12-03848-t003:** Vitamin and mineral contents in tip leaves, young leaves, intermediate leaves and stems of liang (*Gnetum gnemon* var. *tenerum*).

Compound	Treatments				Nutrition Claim Criteria (Wet) *	
	Tip	Young	Intermediate Leaves	Stem	Good Source	High
Vitamins						
A (Beta-Carotene) (mg/100 g)	1.35 ± 0.06 ^c^ (0.24 ± 0.01 ^c^)	2.35 ± 0.09 ^b^ (0.40 ± 0.02 ^b^)	3.03 ± 0.07 ^a^ (0.51 ± 0.01 ^a^)	2.23 ± 0.08 ^b^ (0.38 ± 0.01 ^b^)	≥0.72	≥1.44
B_9_ (Folic acid) (mg/100 g)	0.0290 ± 0.0013 ^b^ (0.0052 ± 0.0002 ^b^)	0.0375 ± 0.0002 ^a^ (0.0064 ± 0.0000 ^a^)	ND ^c^ (ND ^c^)	ND ^c^ (ND ^c^)	≥0.03	≥0.06
C (Ascorbic acid) (mg/100 g)	5.25 ± 0.05 ^a^ (0.94 ± 0.01 ^a^)	2.93 ± 0.05 ^b^ (0.50 ± 0.01 ^b^)	2.71 ± 0.04 ^c^ (0.46 ± 0.01 ^c^)	ND ^d^ (ND ^d^)	≥9	≥18
Minerals						
Calcium (mg/100 g)	374.11 ± 6.42 ^d^ (67.04 ± 1.15 ^b^)	425.89 ± 16.26 ^b^ (72.7 ± 2.78 ^a^)	450.71 ± 5.37 ^a^ (76.44 ± 0.91 ^a^)	400.18 ± 3.08 ^c^ (68.23 ± 0.53 ^b^)	≥120	≥240
Copper (mg/100 g)	0.56 ± 0.02 ^a^ (0.10 ± 0.00 ^a^)	0.47 ± 0.02 ^b^ (0.08 ± 0.00 ^b^)	0.47 ± 0.01 ^b^ (0.08 ± 0.00 ^b^)	0.47 ± 0.00 ^b^ (0.08 ± 0.00 ^b^)	≥0.3	≥0.6
Iron (mg/100 g)	4.46 ± 0.20 ^a^ (0.80 ± 0.04 ^a^)	3.63 ± 0.05 ^b^ (0.62 ± 0.01 ^b^)	3.63 ± 0.04 ^b^ (0.61 ± 0.01 ^b^)	1.94 ± 0.05 ^c^ (0.3 ± 0.013 ^c^)	≥2.25	≥4.5
Magnesium (mg/100 g)	196.76 ± 0.60 ^a^ (35.26 ± 0.11 ^a^)	197.01 ± 8.23 ^a^ (33.63 ± 1.41 ^ab^)	193.1 ± 0.27 ^a^ (32.75 ± 0.05 ^b^)	124.52 ± 6.21 ^b^ (21.23 ± 1.06 ^c^)	≥52.5	≥105
Zinc (mg/100 g)	4.58 ± 0.02 ^a^ (0.82 ± 0.00 ^a^)	3.51 ± 0.16 ^b^ (0.60 ± 0.03 ^b^)	3.18 ± 0.07 ^c^ (0.54 ± 0.01 ^c^)	2.23 ± 0.07 ^d^ (0.38 ± 0.01 ^d^)	≥2.25	≥4.5

Values in parentheses are wet basis; a–d means different lowercase letters indicate significant differences between treatments (*p* < 0.05). Source (*) Ministry of Public Health [23].

**Table 4 foods-12-03848-t004:** The total coliform count of liang (*Gnetum gnemon* var. *tenerum*) leaves after storage at 4 °C for 12 days (MPN/g).

Time/Condition	NWS	NWNS	WS	WNS
D0	>1100	>1100	>1100	>1100
D4	>1100	>1100	240	240
D8	1100	>1100	<3	150
D12	>1100	>1100	7.4	>1100

NWS means no-washing with stem; NWNS means no-washing without stem; WS means washing with stem; WNS means washing without stem.

**Table 5 foods-12-03848-t005:** *Escherichia coli* contents of liang (*Gnetum gnemon* var. *tenerum*) leaves after storage at 4 °C for 12 days (MPN/g).

Time/Condition	NWS	NWNS	WS	WNS
D0	93	93	93	93
D4	150	150	<3	3.6
D8	43	>1100	<3	150
D12	43	460	<3	23

NWS means no-washing with stem; NWNS means no-washing without stem; WS means washing with stem; WNS means washing without stem.

## Data Availability

All data are contained within the article and Appendix A.

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
