# Peer review of "Nutritional Content and Microbial Load of Fresh Liang, Gnetum gnemon var. tenerum Leaves"

_foods, 2023, doi:10.3390/foods12203848_

Round 1
Reviewer 1 Report
The article is interesting, however some minor recommendations and comments are suggested.
The introduction could be structured in paragraphs for better organization of ideas.
Line 107-109. Indicate that this method belongs to the determination of moisture or what method is this?
Line 114. Was the Kjeldahl method used in this methodology? please indicate.
Line 129. Was the Soxhlet method used in this methodology? please indicate.
Line 287-288. In this part “). Highest levels of protein, fat, and ash were 287 found in the tip leaves and decreased with age.” Why? Discuss more about it.
Lines 337-339. In this part “Interestingly, folic acid was only found in the tip and young leaves and was absent in the intermediate and old leaves.” Why? Discuss more about it.
The conclusion could be completed with some future perspectives on this.
Author Response
The introduction could be structured in paragraphs for better organization of ideas.
-I already edited introduction as suggestion.
Line 107-109. Indicate that this method belongs to the determination of moisture or what method is this?
-I already edited line 107-109 by adding heading moisture content
Line 114. Was the Kjeldahl method used in this methodology? please indicate.
-I already indicated as followed “Protein determination was analysed by Kjedahl method according to AOAC [8] method 981.10”
Line 129. Was the Soxhlet method used in this methodology? please indicate
-It’s not Soxhlet method it is acid hydrolysis based on AOAC (2019).
Line 287-288. In this part “Highest levels of protein, fat, and ash were 287 found in the tip leaves and decreased with age.” Why? Discuss more about it.
-The reason was explained in line 356-362.
Lines 337-339. In this part “Interestingly, folic acid was only found in the tip and young leaves and was absent in the intermediate and old leaves.” Why? Discuss more about it.
-I already added more discussion in line 400-408.
The conclusion could be completed with some future perspectives on this.
-I already added future perspectives in conclusion.
Reviewer 2 Report
The authors did a good work from an experimental point of view, and I recommend the article for publication after some major revisions.
More specific:
L73: Chemicals and reagents sub-section is missing.
L75: You should also mention the country.
L85: 100 ppm is a high chlorine concentration! Are you sure? Coughing and vomiting may occur at 30 ppm and lung damage at 60 ppm.
Figure 1: Improve resolution!
L103: Make it cleaner. This sub-section needs revision. Separate the methods.
L194: Please provide more details on amino acids analysis.
L204: Which vitamin? Did you analyze vitamins A, B9, and C with the same protocol of analysis?
All equipment used needs more details. Model, company, town, country missing.
L279: More details needed for statistical analysis.
Figures 6 and 7 need a modified format. Give different colors to the columns.
In the supplementary file, you can try changing the format axis ranging from 0.990 to 1.000.
Extensive editing of English language required.
Author Response
L73: Chemicals and reagents sub-section is missing.
-I already added chemical and reagents sub-section.
L75: You should also mention the country.
-I already added “Thailand”.
L85: 100 ppm is a high chlorine concentration! Are you sure? Coughing and vomiting may occur at 30 ppm and lung damage at 60 ppm.
-I already added more information because 100 ppm chlorine concentration in washing water was allowed but chlorine residue in food must lower than 1 mg/kg.
Figure 1: Improve resolution!
-I already improve resolution of figure 1.
L103: Make it cleaner. This sub-section needs revision. Separate the methods.
-I already edited this subsection.
L194: Please provide more details on amino acids analysis.
-I already provided more concise details on amino acid analysis.
L204: Which vitamin? Did you analyze vitamins A, B9, and C with the same protocol of analysis?
- I already added information and edited all method for determined vitamin.
All equipment used needs more details. Model, company, town, country missing.
-I already added details of equipment.
L279: More details needed for statistical analysis.
-I already added more information.
Figures 6 and 7 need a modified format. Give different colors to the columns.
-I already edited color of figures 6 and 7.
In the supplementary file, you can try changing the format axis ranging from 0.990 to 1.000.
-I already edited figure in supplementary file as your suggestion.
Extensive editing of English language required.

Reviewer 3 Report
The authors have studied nutritional content and microbial load of fresh Liang leaves, a green vegetable described as widely consumed in Southern Thailand. The leaves of Liang appear with multiple nutritional benefits, in the same time they are susceptible to the microbial infection, as a result of both, growing conditions and the multiple maneuvers and treatments from the farmer to the end user.
Overall, knowing the content and dynamic of the nutritional compounds along the stage of growth, and also from the moment of harvesting and storage till de moment of sale, in the same time knowing the critical points of microbial attack and the ways of avoiding or diminishing the microbial load, is of real scientific and practical importance.
Point by point: the introduction is very clear; the part presenting materials and methods is appropriate; the results part is clear and concise; discussions are accurate, and conclusions in accordance with the results. Tables and Figures are clear and easy to understand. Also, the author presented other utile statistical data, aspects related to the legislation and FDA, therefore the study is very good.
All these aspects will ensure a high number of citations in the future.
As a weak point, but which does not reduce the value of the work, the authors could have offered some alternatives for washing composition, therefore other than chlorinated water, even if they are not recommended or used in the current practice.
Author Response
As a weak point, but which does not reduce the value of the work, the authors could have offered some alternatives for washing composition, therefore other than chlorinated water, even if they are not recommended or used in the current practice.
-I already added more information about other washing agent and method at line 585-588.
Round 2
Reviewer 2 Report
The paper has been revised according to the suggestions and criticisms of the reviewers. In this revised version, the paper has improved its quality and I recommend the article for publication.